# Risk and Protective Factors of Mental Health Conditions: Impact of Employment, Deprivation and Social Relationships

**DOI:** 10.3390/ijerph19116781

**Published:** 2022-06-01

**Authors:** Beatriz Oliveros, Esteban Agulló-Tomás, Luis-Javier Márquez-Álvarez

**Affiliations:** 1Social Education, Faculty Padre Ossó, University of Oviedo, 33008 Oviedo, Spain; 2Department of Psychology, University of Oviedo, 33003 Oviedo, Spain; estomas@uniovi.es; 3Occupational Therapy, Faculty Padre Ossó, University of Oviedo, 33003 Oviedo, Spain; luisjavier@facultadpadreosso.es

**Keywords:** mental health, risk factor, protective factor, exclusion, deprivation, job insecurity

## Abstract

This study looks into the relationship between mental health and social exclusion scenarios, paying special attention to employment-related factors. Previous studies have shown the relationship between mental health, social exclusion and poverty. For this study, authors have used data from the VIII Report on social development and exclusion in Spain, with a sample of 11,655 households. The SPSS Statistics programme was used for statistical analysis. Several factors that could pose a risk or be a protection for the presence of mental health conditions were designed. By means of a binary logistic regression the impact of these factors on mental health issues was scored. The results show that a deteriorated social network and a negative interpretation of reality are the most influential factors related to the presence of mental health conditions in a given household. On the contrary, positive social relationships protect households and function as a support when mental health conditions are already present. Thus, the support of positive and committed social relationships is a key element to protect the mental health of households.

## 1. Introduction

Mental health, according to World Health Organization (WHO), is “a state of well-being in which an individual realizes his or her own abilities, can cope with the normal stresses of life, can work productively and is able to make a contribution to his or her community” [1]. It is a key aspect in peoples’ lives, but also necessary for economic growth and social development. This concept is directly related to quality of life (QoL) and well-being. Its decrease or deficit results in health issues, deprivation and relationship problems but also in problems concerning education, employability and job performance [2]. Data show that individuals suffering from mental health problems also indicate a low employability rate which may lead to a greater poverty and social exclusion risk. Mental health in developed countries accounts for between a third and half of long-term medical leave [3].

Since 2008, the WHO has studied the impact on health as well as the impact on mental health on the basis of the model of Social Determinants of Health which are defined as the conditions in which people are born, grow, work, live, and age, and the wider set of forces and systems shaping the conditions of daily life [4]. Ten social determinants of health have been identified, among which: the social gradient (i.e., the phenomenon whereby people who are less advantaged in terms of socioeconomic position have worse health than those who are more advantaged), stress, early childhood, social exclusion, employment, unemployment, social support, addictions, food and transport [5,6].

As of 2010 there has been an increase in scientific production regarding the social and occupational impact of mental health [7], but knowledge gaps remain in many aspects. The greater interest in generating knowledge related to this topic may be linked to the current high impact of mental health on social costs, whereby the prevalence rate of mental disorders amounts to 38.2% [8,9,10,11]. Mental health problems may be both a result of and a risk factor for unemployment [12]. Risk ratios of mental health issues among the unemployed increase between 1.5 to 3.5 times more than among the employed, whereby the greater the unemployment the more severe the condition [13,14,15,16].

According to 2012 data, the percentage of employed individuals with mental health conditions is greater than that of unemployed individuals with mental health conditions. Although data vary depending on the country of reference in Europe, approximately 50% of individuals with severe mental health disorders have a job; this data applies up until 2021 [3,17]. However, there is still a large number of individuals who cannot find a job and are unemployed, which exacerbates the stigmatisation processes faced by these individuals due to the fact of having a mental ill-health issues [18].

In terms of the working conditions of these people, there is a high percentage of workers with different mental health conditions working part-time. While there is a group of workers who had no choice over their work schedule, the other group have used this reduction of working hours to balance the demands resulting from their mental health condition. In this sense, a flexible schedule (flexitime) seems to be a key aspect for people to remain in the labour market [17].

Table 1 shows an analysis of the National Health Survey in Spain (Encuesta Nacional de Salud en España-ENSE) [19] in relation to the presence of mental health disorders based on the labour situation informed by the respondent. The profile of people with the worst mental health conditions is that of individuals who, due to disability, cannot access the labour market and stand out with a value above 4; they are followed by unemployed people who only do household chores and those in early retirement/retirement.

In recent years a lot of research has been done on the impact of unemployment on the physical and mental health of people [2,18,20,21,22]. In this regard, it is interesting to highlight the contributions by Espino [20] related to the effects of unemployment on mental health. The author states that unemployment is a relevant risk factor for the emergence of mental health problems: anxiety, insomnia, depression and dissocial and self-destructive behaviours.

The current situation due to COVID-19 has resulted in an even greater impact on health and economics. The work environment has changed significantly and many jobs have been lost, in particular jobs performed by women [23,24]. The extraordinary working conditions generated to respond to the exceptional situation resulting from the pandemic has widened the gender gap linked to employment. Other people have experienced how their own home has become their workplace amidst a technological revolution that has adapted working conditions to new tasks. A range of positive benefits are associated with teleworking, including improved family and work integration, reductions in fatigue and improved productivity [25]. “However, the blurring of physical and organisational boundaries between work and home can also negatively impact an individual’s mental and physical health due to extended hours, lack of or unclear delineation between work and home, and limited support from organisations” [26].

In this new scenario, it is necessary to establish more evidence on the different risk and protective factors that may strengthen or weaken the mental health of people. To such an end, the following terms should be conceptualised: (a) Protective factor: characteristic at the biological, psychological, family, or community (including peers and culture) level that is associated with a lower likelihood of problem outcomes or that reduces the negative impact of a risk factor on problem outcomes; (b) Risk factor: characteristic, condition, or behaviour that increases the likelihood of getting a disease or injury [27,28].

This study aims to identify the factors that have an impact and are a protection or a risk for the onset of mental health conditions in households and their impact on the labour market.

## 2. Materials and Methods

The design of the study was based on a cross-sectional, descriptive, analytical and exploratory approach on the basis of a country-wide sample (Spain). This research follows the requirements and protocols of the Ethics Committee of Oviedo University, where it was performed, as well as all the ethical demands and recommendations included in the Ethical Principles of Psychologists and Code of Conduct of the APA. This study was conducted in accordance with the Declaration of Helsinki. Data have been provided by the Foundation for the Promotion of Social Studies and Applied Sociology (FOESSA).

### 2.1. Participants

The sample used in this study is the one used for the VII Report on social development and exclusion in Spain [29]. The survey aims to quantify and analyse living conditions and social exclusion in Spain, obtaining representative data both on a national level and on the level of autonomous communities and territories.

The report has been drafted on the basis of a sample including 11,655 households. For the general sample the initial maximum error is set at 2%, although the final error will be much lower due to the specific samples. With these sizes an error level of ±5%, is guaranteed, with a confidence interval of 95% for the population with signs of inclusion, and the same error level for all households in general. The sampling has been random and stratified. The total sample starts from an approach of statistical representativeness of the national population and, thus, each household has been weighted over the total value depending on their city, community, gender or age. This method guarantees that results have a high level of representativeness with respect to the total population.

### 2.2. Instruments

The set of EINSFOESSA (Encuesta sobre Integración y Necesidades Sociales) exclusion indicators has been used since 2007 and has been employed in the four editions prior to the survey [29,30,31,32]. This survey includes information focused on three spheres related to social exclusion: (a) economic sphere; (b) political sphere or sphere of access to citizen’s rights; (c) social and relational sphere. These spheres are broken down into eight dimensions: (a) Employment; (b) Consumption; (c) Politics; (d) Education; (e) Health; (f) Housing; (g) Social conflict; and (h) Social isolation. Independent variables collected add data about: (a) housing; (b) gender; (c) relationship with the main breadwinner; (d) civil status; (e) legal situation.

### 2.3. Procedures and Analysis

The information was collected in person. The selection of households is carried out by randomised routes within the section, defined by the INE (Instituto Nacional de Estadística, Spain) street maps. Two questionnaires were used to collect information: one was used as a screening questionnaire to classify the sample; the other, main one, collected all the information. When the interviewer arrived at a household firstly he/she used the filter questionnaire to identify the main socioeconomic characteristics of the household. On the basis of the outcomes of this questionnaire, the household was classified in a given category and the main questionnaire was completed, on the basis of the necessary quotas for that area.

All FOESSA Foundation researchers used the questionnaire of the VII Report on social development and exclusion in Spain [29]. The SPSS programme was used for sample analysis (v23.0). All variables of interest were included in a database available to the research team.

It was decided that the binary logistic regression model was the most appropriate in line with the research approach and the data handling on the basis of the questionnaire used, among other reasons because the selected dependent variable is binary in nature. In accordance with the objective of this research, this selection is justified on two grounds: (a) the sample is large enough to provide reliable data with a significance level lower than 0.05; (b) the 11,655 households of the sample represent a number large enough to obtain adequate reliable margins.

The variables have been obtained through the binary logistic regression model to build two models that explain the existing relationship between them for the purpose of risk factors (called risk model) or protective factors (called protection model) for mental health in the household of each interviewed family. For clarification purposes, results are shown in Odds ratios (OR) measures. OR that are greater than 1 shall indicate that the event is more likely to occur as the predictor increases. OR that are less than 1 indicate that the event is less likely to occur as the predictor increases. This is translated interpretatively as protective factors on the protective model and risk factors on the risk model.

To perform the selected analyses two dependent variables were used to create such models: (a) the presence of a mental condition in the household for the model of risk factors; (b) the absence of a mental health condition in the household for the model of protective factors. The independent categorisation of these models allows us to calculate the actual value and the implications of the different independent variables. By following this structure, each variable provides a different weight to the presence or absence of a mental health disorder and allows for a better understanding of the phenomena studied.

## 3. Results

After analysing, through the binary logistic regression model, the relationships between the different variables, two models have been elaborated. Both models have taken into account a sufficient number of response variables that may explain the presence of mental conditions in households. The model including risk factors started with 111 variables, until the final 29 variables were obtained. The model including protective factors started with 30 variables until the final 8 variables were obtained.

### 3.1. Risk Model

Table 2 shows the variables included in the risk model, their significance level and the corresponding OR which will be analysed in the following items.

It is worth mentioning that all spheres are represented through some of their dimensions, although some weigh more than others; for example, deprivation in the economic sphere, or the social sphere that is represented in all dimensions.

Results show that financial capacity is one of the greatest influences on mental health conditions in Spain. Financial difficulties are related to the lack of sufficient income in the household to enjoy a decent standard of living that allows its members to access basic services. Many of these services are linked to health.

Table 2 shows that, due to the failure to cover certain expenses related to health, the likelihood and risk of onset of a mental health condition in the household increases: the fact of not buying medication due to financial difficulties multiplies by 2.8 the possibility that a member of the household will develop a mental health condition.

From a health standpoint, the table shows that the three variables included in the model and related to this sphere have to do with the existence in the household of people with chronic conditions, dependency or limitations to perform basic daily activities autonomously. In all of them, ORs are close to 1.5, that is, they multiply by 1.5 the likelihood of there being a mental health condition in the household.

Risk behaviours are, by definition, a risk for households. But this risk is the most significant and should be taken into account by social policies as the ORs are much higher than in any other of the fields analysed in this study.

Family units with prior criminal records multiply by 2.4 the potential existence of mental health disorders. lients of services catering to people with disabilities, drug addicts, the elderly, homeless, or women are 4.8 times more likely to experience mental health issues. People experiencing alcohol, drug, or gambling addiction also show a very high OR ranked between 3 and 5.8.

From a gender standpoint, worth noting is the discrimination suffered by some women for the very fact of being women (with a 1.7 OR) and psychological abuse with a 4.6 OR.

Finally, data show how a negative perception, from the point of view of a worsening of the personal situation or the lack of hope, poses a risk to mental health. Vital perception variables in Table 2 are related to the lack of life satisfaction and the perception of no improvement. In all cases, the OR is positive and greater than 1, reaching almost 2 in all the households in which the respondent did not feel satisfied with her/his life.

### 3.2. Protection Model

Table 3 shows the variables included in the protection model, their significance level and the corresponding OR to be analysed in the following items.

In terms of the protection model, the first aspect to be highlighted is that the number of variables and the impacted spheres and dimensions is much lower than in the risk model. However, all the spheres are present though the economic and political spheres lose strength in comparison to the social sphere, which is strongly represented.

In terms of deprivation, the simple fact of having a sufficient income level to maintain the household at an optimal temperature protects households against mental conditions: more specifically, protection level amounts to 1.4.

From a health standpoint, the table shows that those without any severe health conditions are more protected against mental health conditions, in particular, almost twofold in comparison to households with a chronic condition.

Table 3 shows that four of the eight variables included in the protection model are related to social relationships. Having frequent relationships with friends multiplies by 1.33 the protection against mental conditions. Furthermore, frequent relationships with neighbours provide a 1.27 protection level. But not only is the frequency of relationships assessed but also their quality. The greatest protection level is indeed found in this quality. Having good relationships with non-cohabitant relatives multiplies protection by almost 2.1, whereas having good relationships with neighbours multiplies it by 1.6.

Results show that a positive perceived quality of life is also a strong protector of mental health. The respondents who think that their household income level will not make them need financial support in the following 12 months multiply protection by 1.2. The respondents who express satisfaction with their lives multiply protection by 1.7.

## 4. Discussion

This study aimed to identify the protective and risk factors that have an impact for the onset of mental health conditions in households and their impact on the labour market. Some recent studies had already revealed the scarce protective role of the current employment system in the prevention of exclusion scenarios and mental health conditions [33]. As the results have shown, none of the models attach a decisive role to situations of employment, unemployment, precarious employment or job insecurity: they are all absent.

On the other hand, neither unemployment (short or long-term) nor employment when the individual is already in an exclusion scenario, nor irregular employment, stand out as variables with significance in their relationship with the presence of mental health disorders.

Regarding the existence of employment in the model, only one variable refers to this field: households in which one member has applied for unemployment benefits multiplies by 1.3 the presence of mental health disorders. In this regard, one could assess the deficient coverage of the benefits, contributory and, to a greater extent non-contributory ones, to cover household needs. But still, it could seem that the current unemployment coverage, contributory in nature, does protect them from mental health conditions. Unemployment may not be linked to issues such as social exclusion or social vulnerability. This is not the case of poverty, as in the case of poor workers [33,34,35].

This fact links directly to the need for a guaranteed minimum income. In this regard, Fernández [36] states that the income guarantee system is a support for citizens to maintain a decent living standard and affirms that it impacts on poverty reduction as well as on maintaining well-being in the household. The resulting data allows us to show that an adequate economic coverage could prevent a greater risk posed by the presence of mental health disorders. Evidence that emerges from the analysis shows a relevant impact of material deprivation on the risk of mental conditions. Austerity policies implemented on occasion of the Global Financial Crisis in 2008 as well as the scarce political interest in introducing a minimum, sufficient and adequate income in the different territories as a commitment to the social agenda expose families to a risk of a mental health conditions [29,30,32].

In general, situations of severe illness and dependency are closely linked to the presence of mental health disorders in households and their absence. Situations of good health and well-being function as a clear protective factor. Situations of deteriorated health usually highlight the need for the presence of an “informal care-giver”, defined by the Institute of the Elderly and Social Services (Instituto de Mayores y Servicios Sociales-IMSERSO) as the care provided to individuals in a situation of dependency at home, by relatives or persons around him/her who are not linked to a professional care service [37]. The risk of the presence of mental health conditions in these caregivers is greater. This is emphasised by López et al. [38], who states that some processes such as anxiety, depression, sleep disturbance, apathy and irritability are more typical of informal caregivers in comparison to the rest of the population. This reality has intensified during the pandemic as people taken care of had limitations to access specialised resources due to lockdowns. The pandemic has had a negative impact on the mental health of family caregivers, especially affected by loneliness and excessive care-related responsibility [39].

Social relationships generate solidarity networks that are real social resources, but they also generate processes of significance and identity, no less important, in the symbolic dimension of social integration [40].

The working hypotheses had included the possibility of the fact that the absence of healthy and committed social relationships could be one of the aspects that most influenced the emergence of risks regarding the presence of mental health conditions. Such has been the case: households with bad relationships with other relatives or neighbours are twice as exposed to the presence of a mental health disorder. This agrees with Raynor’s et al. findings, where residents of group households characterised by pre-existing precariousness were vulnerable to negative mental health effects [41], or Gan et al. findings, where neighbourhood cohesion plays an important role in the mental health of residents [42]. Also Subirats et al. mention the importance of socialising and relational factors that allow for the genesis of social and community ties promoting social inclusion processes [43].

One curious point is how variables that refer to mutual support (people who have nobody to help them or who are not of help to others), although significant in the model, do not expose households to the presence of mental health disorders but rather are a kind of protective framework that has not been looked into. This may be due to the deterioration of family support networks (above all the ones related to the extended family) due to the crisis. Many research projects performed in recent years mention social media “burnout”, above all the extended family [44,45,46,47,48,49,50].

In the section on social anomie and conflict the variables selected are related to situations and behaviours that are a disruptive element in the household. One way to express social exclusion scenarios has to do with relationships that show a perverse dimension or are manifested in behaviours with wide social rejection [40]. In this case the problem does not lie in the absence of social ties but rather in the fact that the existing ties place the involved individuals outside of society as a whole.

At this point it should be mentioned that one of the studied mental health categories is related to disorders linked to addictions, namely, dual pathologies that are very frequent in people in a situation of exclusion with addiction issues. In this case, the data linked to consumption in the presence of a mental health disorder is worsened, as stated by Torrens [51] who affirms that these “dual” patients or with a psychiatric comorbidity are frequent and show more severity from a clinical as well as a social perspective, as they do not only have one type of disorder (addictive or any other psychiatric condition). In any event, it cannot be confirmed whether these data refer exclusively to the person with an addiction and mental issues or to another person who shares the household.

Physical abuse has not been significant and has been withdrawn from the models. In relation to this, research on gender violence in the household has highlighted the impact of reiterated and prolonged abuse on mental health. For example, Amor et al. stated that only a small percentage of victims had a prior psychiatric record, mainly emotional issues like anxiety and depression. These are, therefore, mentally balanced women who currently suffer from psychological disorders as a result of a context of chronic abuse [52].

The perception regarding the subjective experience of reality is paramount in that important dimension of life called “happiness”. Some authors have focused on this idea and have designed scales to measure life satisfaction [53]. This satisfaction can be understood as a comparison between global life circumstances and other imposed standards [54]. Data show how a positive vision of the present and a positive outlook on the future function as a protective element. On the contrary, negative future outlooks are an important risk for the onset of mental health conditions.

Finally, it may be noted that we have found two important limitations in this study. Firstly, the fact that the questionnaire asks about the presence of mental health conditions in the household, without specifying the member of the household being referred to. In future it would be necessary to define who this/these person/persons are to perform the analysis in greater depth. The second limitation refers to the interpretation of the outcomes obtained in those variables that, although significant, function conversely.

Also, the cultural implications of the findings do not validate them for societies other than Spain. Even with the selection of households, it could not be overlooked that mental health has a cultural component that should be referred. Our model tried to expose relevant variables that other researchers could contrast with their current cultures. Because of this, the thorough study of these variables could be a future line of work in upcoming research studies.

## 5. Conclusions

This study has shown the impact of the different variables of interest on mental health in households. Such variables may be understood jointly, creating two models to facilitate their understanding.

Results have shown how certain variables (above all, those related to the subjective perception of personal reality and conflictive social relationships) have a high impact on the presence of mental disorders in people. Thus, in the design of social inclusion processes based on mental health, the following determining factors should be taken into account:

Risk behaviours are the most significant risk factors. It should be taken into account by social policies as the OR for this area are much higher than the OR in any other field analysed in this study.

Employment is not a guarantor of inclusion or protection of mental health. The fact of insisting on inclusion processes linked to employment does not lead to the improvement of individuals’ health, but rather it exposes people to precarious situations resulting in a more severe and intense mental health illness and suffering.

The lack of adequate and sufficient income does impact on the mental health of people. Therefore, a guaranteed minimum income to cover basic needs is a priority to be taken into account in social inclusion models. In this sense, it is also necessary to implement housing policies related to access to regular supplies. Support for families, facilitating their access to necessary supplies to maintain the household, is a way to protect their health, especially their mental health.

Healthcare, from a broad healthcare and health approach, is one of the most important elements to adjust mental health to life circumstances. Prevention programmes that prioritise primary health care as the first step to find out about a person’s state of health are paramount. A solid budget will allow for public and high-quality health services focused on the strengthening of primary care and mental health.

A stable, healthy and committed network of relationships is highly protective in the presence of mental health conditions. More attention should be paid to the different relational dynamics developed by people with mental health conditions, reinforcing those subjected to greater stress. Public policies should take into account initiatives that provide social spaces and consolidate social and community ties.

The model provided enables rethinking the relationship between health and mental health on the basis of social determinants of health. It also allows us to establish priorities in the fields of mental health protection from a social and health care approach. It allows us to address mental health in a multidisciplinary way, focusing not only on a purely biologic approach but rather on the impact of the social sphere and how it determines lifestyles.

It is necessary to create more lines of research into the influence of these determinants and look into their evolution with their different and changing social situations.

## Figures and Tables

**Table 1 ijerph-19-06781-t001:** Prevalence (%) of mental health issues related to economic activity. Source: Encuesta Nacional de Salud en España (ENSE) 2017 [19].

	Depression	Anxiety	Other
Active workers	3.08	4.42	0.50
Unemployment	7.90	9.44	1.39
Retirement/Early retirement	11.14	8.47	3.91
Studying	1.57	2.44	1.54
Work disability	30.06	27.35	22.94
Housework	12.14	9.77	1.45

**Table 2 ijerph-19-06781-t002:** Variables included in the risk model, significance level (*p*) and corresponding Odds Ratio (OR).

Field	Description	*p*	OR
Employment	Have applied for unemployment benefits	*0.004*	1.298
Deprivation	Do not purchase drugs due to financial difficulties	*0.000*	2.805
	Do not attend therapy due to financial difficulties	*0.000*	2.413
	Need medical care but they are still on the waiting list	*0.013*	1.773
	Loss of social relationships due to financial difficulties	*0.011*	1.326
	Need to ask a relative or friend for help	*0.009*	1.290
	Housing payment delays	*0.002*	0.587
	Cannot afford hospitalisation costs	*0.000*	0.360
	Treatment withdrawal due to financial difficulties	*0.000*	0.185
Political participation	Do not take part in elections	*0.005*	0.596
Health	Household with severe disorders	*0.000*	1.682
	Household with disabilities	*0.002*	1.555
	Households with situations of dependency	*0.001*	1.458
Social relationships	Poor relationships with other relatives	*0.006*	1.906
	No relationships with neighbours	*0.001*	1.645
	Lacking a support network	*0.000*	0.572
	They do not provide support to other people	*0.000*	0.548
Risk behaviours	People with drug addictions in the last ten years	*0.000*	5.810
	Households with members in institutions (mental health, nursing homes, etc.)	*0.000*	4.869
	Households with psychological abuses in the last ten years	*0.000*	4.611
	Households with gambling issues in the last ten years	*0.001*	3.838
	Households with alcohol abuse in the last ten years	*0.000*	2.934
	Households with criminal records in the last ten years	*0.015*	2.404
	Gender discrimination (women)	*0.000*	1.710
Vital perception	Households that manifest life dissatisfaction	*0.000*	1.868
	Households self-described as poor	*0.001*	1.469
	Households that do not perceive the economic recovery	*0.009*	1.282
	Households whose living standard has worsened in the last ten years	*0.037*	1.184

**Table 3 ijerph-19-06781-t003:** Variables included in the protection model, significance level (*p*) and corresponding Odds Ratio (OR).

Field	Description	*p*	OR
Deprivation	Can keep their home warm	*0.000*	1.366
Health	Households without severe diseases	*0.000*	1.990
Social relationships	Frequent contact with friends	*0.000*	1.333
	Frequent contact with neighbours	*0.005*	1.247
	Good relationships with other relatives	*0.000*	2.062
	Good relationships with neighbours	*0.000*	1.561
Perceived quality of life	Households that think they will not need financial assistance in the following 12 months	*0.007*	1.253
	Households in which the interviewee is satisfied with his/her life	*0.000*	1.752

## Data Availability

Additional data can be accessed in FOESSA foundation webpage.

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
