# Peer review of "Risk and Protective Factors of Mental Health Conditions: Impact of Employment, Deprivation and Social Relationships"

_ijerph, 2022, doi:10.3390/ijerph19116781_

Round 1

Reviewer 1 Report

Thank you very much for give me the opportunity to review this interesting paper titled

Risk and protection factors of mental health conditions: Up to what extent is employment a determinant?

  1. Altogether it is an interesting study with a complex and very complete methodology, quite well written and with an interesting discussion section and conclusions. Say this, in my modest opinion it would be convenient to help the reader a little bit more to follow the authors. Perhaps it could be useful, for example, if you recalled the objective of your work at the beginning of your discussion section.
  2. I am more familiar with the expression p in italics to indicate statistical significance (for example in table 2 your whole results are significant).
  3. Please carefully check the format (e. g. on line 31, you might have an extra point and followed between performance and the quotation marks and in lines 37, 181, 183, 256, 281 maybe a space is missing).
  4. Please check the phrase “The profile of people with greater 73 mental health conditions is that of individuals who, due to disability, cannot access the labour market and stand out with a value above 4; they are followed by unemployed people who only do household chores and those in early retirement/retirement.” As you shown in table 1 such people group exhibited the greater prevalence of depression, anxiety, and other health problems, and therefore it seems that the worst mental health conditions.
  5. Please check the references section, it does not seem to completely follow the format indicated by the journal. 

Author Response

To the International Journal of Environmental Research and Public Health editor,

Thank you for giving me the opportunity to submit a revised draft of my manuscript titled “Risk and protection factors of mental health conditions: Impact of employment, deprivation and social relationships” (retitled) to the International Journal of Environmental Research and Public Health.

We appreciate the time and effort that you and the reviewers have dedicated to providing your valuable feedback on our manuscript. We are grateful to the reviewers for their insightful comments on the paper.

Here is a point-by-point response to the reviewers’ comments and concerns. Also, we attached the manuscript with the subsequent corrections.

Thank you again for your support.

Reviewer 1

R2-1. Thank you very much for give me the opportunity to review this interesting paper titled Risk and protection factors of mental health conditions: Up to what extent is employment a determinant?

Altogether it is an interesting study with a complex and very complete methodology, quite well written and with an interesting discussion section and conclusions. Say this, in my modest opinion it would be convenient to help the reader a little bit more to follow the authors. Perhaps it could be useful, for example, if you recalled the objective of your work at the beginning of your discussion section.

Author’s response 1: We thank the reviewer for taking the time to assess our manuscript and the given feedback. The comments provided valuable insights to refine its contents and analysis.

As suggested by the reviewer we changed the objective of the research at the beginning of the discussion. Now reads as follows:

“This study aimed to identify the protective and risk factors that have an impact for the onset of mental health conditions in households and their impact on the labour market.”

R2-2. I am more familiar with the expression p in italics to indicate statistical significance (for example in table 2 your whole results are significant).

Author’s response 2: Following the suggestion by the reviewer, Table 2 and Table 4 columns for p values have been changed to italics.

R2-3. Please carefully check the format (e. g. on line 31, you might have an extra point and followed between performance and the quotation marks and in lines 37, 181, 183, 256, 281 maybe a space is missing).

Author’s response 3: We would like to thank the reviewer for pointing this out. We double checked the actual version of the manuscript with language experts to avoid these mistakes in its actual form.

R2-4. Please check the phrase “The profile of people with greater mental health conditions is that of individuals who, due to disability, cannot access the labour market and stand out with a value above 4; they are followed by unemployed people who only do household chores and those in early retirement/retirement.” As you shown in table 1 such people group exhibited the greater prevalence of depression, anxiety, and other health problems, and therefore it seems that the worst mental health conditions.

Author’s response 4: We have fixed the error as we think it must be a problem with the translation (“greater” as in “greater volume”, not “greater health”). The idea we wanted to share was the one expressed by the reviewer, so the phrase was changed as follows:

“The profile of people with the worst mental health conditions is that of individuals who, due to disability, cannot access the labor market and stand out with a value above 4.”

R2-5. Please check the references section, it does not seem to completely follow the format indicated by the journal.

Author’s response 5: We agree that there was a problem with the references. We reviewed the format of all references to adjust them to the journal standards.

Reviewer 2 Report

Comments to the authors:

The reviewer keenly examined the submitted paper entitled “Risk and protection factors of mental health conditions: Up to what extent is employment a determinant?” He believes that the paper was composed well and may contribute to the literature on mental health studies. This study focuses more on the role of positive social relationships on mental health conditions than the role of social-economic status. However, the title of the submitted paper insinuates that it examined the effect of employment status on mental health conditions rather than the effect of social relationships. Since the title of the submitted paper might mislead the reader concerning its primary findings, the reviewer recommends that the authors change the title to express them appropriately. Moreover, the reviewer has commented on the statistical methods adopted in the study and hopes that they will help improve the paper.

Initially, the reviewer questions the generalization of study findings across industrialized societies beyond the social contexts of Spanish ones. Since Spanish societies are known for respecting familiar bonds, the reviewer believes that the study findings might depend on their social contexts. If the authors assert that their findings are valid for societies other than Spain, it necessitates careful examination. Nevertheless, this is not argued sufficiently in this paper.

Furthermore, the reviewer feels that the statistical model deployed in this study has excessive independent variables. The authors obtained sufficient information about the psychological mechanisms of mental health conditions using many independent variables. However, when independent variables are correlated, the estimated values obtained from (binary logistic) regression models become unstable, leading researchers to underestimate the effects of relevant variables on a dependent variable. The reviewer recommends using structural equation modeling (SEM) to avoid such issues. Since SEM assumes some latent factors among independent variables, the reviewer considers it more suitable to analyze the data based on the research design. At least, the reviewer thinks that the authors needed to explain why they used binary logistic regression model in the study, instead of other appropriate statistical models.

Finally, the analytical results tables of the binary regression models (Table 2-4) only show the odds ratios with the p-values of the independent variables and not the indexes showing the goodness of fit to the data (ex. AIC, BIC, etc.). The statistical significance of the independent variables was easy to determine because the sample size was relatively large (N=11,655). Therefore, the statistical significance of independent variables might be uninformative compared to their effect sizes on dependent variables. Comparing the effect sizes of independent variables using the indexes of the goodness of fit will deliver new findings.

Minor point

Line 222-3. “Table 5 shows the variables related to the lack of life satisfaction and the perception of no improvement.” -> The paper does not contain Table 5.

Author Response

To the International Journal of Environmental Research and Public Health editor,

Thank you for giving me the opportunity to submit a revised draft of my manuscript titled “Risk and protection factors of mental health conditions: Impact of employment, deprivation and social relationships” (retitled) to the International Journal of Environmental Research and Public Health.

We appreciate the time and effort that you and the reviewers have dedicated to providing your valuable feedback on our manuscript. We are grateful to the reviewers for their insightful comments on the paper.

Here is a point-by-point response to the reviewers’ comments and concerns. Also, we attached the manuscript with the subsequent corrections.

Thank you again for your support.

Reviewer 2:

R1-1. The reviewer keenly examined the submitted paper entitled “Risk and protection factors of mental health conditions: Up to what extent is employment a determinant?” He believes that the paper was composed well and may contribute to the literature on mental health studies.

Author’s response 1: We thank the reviewer for taking the time to assess our manuscript and the positive response to the theme of the manuscript. The comments provided valuable insights to refine its contents and analysis. In this review, we tried to address the issues raised as best as possible to meet the expectations of the reviewer.

R1-2. This study focuses more on the role of positive social relationships on mental health conditions than the role of social-economic status. However, the title of the submitted paper insinuates that it examined the effect of employment status on mental health conditions rather than the effect of social relationships. Since the title of the submitted paper might mislead the reader concerning its primary findings, the reviewer recommends that the authors change the title to express them appropriately.

Author’s response 2: We thank the reviewer for pointing this out. We have revised the title to match with the aim and purpose of the paper, now reads as: “Risk and protection factors of mental health conditions: Impact of employment, deprivation and social relationships”.

R1-3. Initially, the reviewer questions the generalization of study findings across industrialized societies beyond the social contexts of Spanish ones. Since Spanish societies are known for respecting familiar bonds, the reviewer believes that the study findings might depend on their social contexts. If the authors assert that their findings are valid for societies other than Spain, it necessitates careful examination. Nevertheless, this is not argued sufficiently in this paper.

Author’s response 3: We agree with the reviewer that further elaborating on this point would be helpful. However, because the sample included each household as a different value, relationships between the family or household members did not apply at the final product of the models. For controlling bias, FOESSA database has a weight on each case that relates it to the standard sample of the Spanish population. Even with this, we agree that there is a possibility of not extrapolating these models to outside-Spain population with mental health issues, so the following was added at Discussion:

“Also, the cultural implications of the findings do not validate them for societies other than Spain. Even with the selection of households, it could not be overlooked that mental health has a cultural component that should be referred. Our model tried to expose relevant variables that other researchers could contrast with their current cultures. Because of this, the thorough study of these variables could be a future line of work in upcoming research studies.”

R1-4. Furthermore, the reviewer feels that the statistical model deployed in this study has excessive independent variables. The authors obtained sufficient information about the psychological mechanisms of mental health conditions using many independent variables. However, when independent variables are correlated, the estimated values obtained from (binary logistic) regression models become unstable, leading researchers to underestimate the effects of relevant variables on a dependent variable. The reviewer recommends using structural equation modeling (SEM) to avoid such issues. Since SEM assumes some latent factors among independent variables, the reviewer considers it more suitable to analyze the data based on the research design. At least, the reviewer thinks that the authors needed to explain why they used binary logistic regression model in the study, instead of other appropriate statistical models.

Finally, the analytical results tables of the binary regression models (Table 2-4) only show the odds ratios with the p-values of the independent variables and not the indexes showing the goodness of fit to the data (ex. AIC, BIC, etc.). The statistical significance of the independent variables was easy to determine because the sample size was relatively large (N=11,655). Therefore, the statistical significance of independent variables might be uninformative compared to their effect sizes on dependent variables. Comparing the effect sizes of independent variables using the indexes of the goodness of fit will deliver new findings.

Author’s response 4: We appreciate the reviewer’s insightful suggestion and agree that a SEM would be useful. However, such analysis is beyond the scope of our paper, as our study aims to detect variables. In this point of the research is important to have a high volume of variables because two reasons: a) the possibilities which offer the FOESSA panel; b) the complexity of the analyzed phenomenon, which is multidetermined [5,6].

Because of this complexity, a SEM would not result operative with these variables. To eliminate variables would not be faithful to the aim of the manuscript, because it could result in a loss of information.

Starting this point, we or other researchers could be in disposition of take advantage of these results to plan future equation models of a more specific nature. The most important thing is that, without the information of this research, the design of future models could be more complex.

Lastly, a very similar method like this has been use in previous research being of great advantage in their respective fields (e.g. Tejero Pérez, A. (2018) “Pobreza laboral en España. Un análisis dinámico”. Revista Internacional de Sociología 76(2):e096. https://doi.org/10.3989/ris.2018.76.2.16.54).

R1-5. Line 222-3. “Table 5 shows the variables related to the lack of life satisfaction and the perception of no improvement.” -> The paper does not contain Table 5.

Author’s response 5: We thank the reviewer for pointing this out; it was changed to Table 3.

Reviewer 3 Report

The authors are to be congratulated for writing a manuscript that I consider acceptable for publication after some revisions.

Four main weaknesses require addressing. First, the cited statistics on page 2 (lines 48 and 56) are outdated: the most recent Spanish statistics are important here. Second, I think you can remove Figure 1 as the statistics are unrelated to Spain.

Third, there needs to be a clear justification for using the Odds Ratio and Risk Ratio as well as a clear definition for each. Statistics in Table 2 need to be numerated as gross totals alongside the p and Odds Ratio columns. This will give the reader a clearer picture of the variables.

Just as important, I think a professional editor will improve this manuscript. The information is there, but the writing is often too wordy. Line 194 is an example,

“Results show that one of the sets that has the greatest impact on the presence of mental health conditions in households is the one linked to financial capacity”.

This can be truncated as:

“Results show that financial capacity is one of the greatest influences on mental health conditions in Spain”.

You may note my highlighted text where this has occurred elsewhere.

-          There is also an excessive use of the phrase ‘in households’. Many of these can be simply deleted. See line 304 for example.

Other than that, other minor corrections are needed.

-          Some words are joined together because of a glitch (e.g., addressthe – line 37), which require separation. I’ve highlighted many in the text, but you’ll need to do a thorough survey of these.

-          Do not use the terms ‘suffering’ or ‘problems’. This implies the person is a passive pawn to their irredeemable fate. Rather, use the words ‘manages’ and ‘issues’. For example, replace ‘individuals suffering from mental health problems’ with ‘people managing mental health issues’. Notice the difference? A person’s agency is emphasized.

-        Acronyms follow the name, which you’ve done correctly sometimes. You don’t need to use acronyms unless used frequently throughout the text (e.g., OR). Just state the name if only used once.   

I've not done all the text that requires reviewing, but here are some:

36 - Delete paragraph. This is information incorrect and unsubstantiated.

123 – delete ‘n=’

124 – Replace ‘%’ symbol with ‘percent’. The symbol is only used in tables or brackets.

134 – EINSFOESSA is not Survey on Social Needs and Integration. Cite the Spanish name if this is so.

146 – INE is not National Statistics Institute. Cite the Spanish name if this is so.

148 – delete (short questionnaire)

192 – The bracketed text should be a standalone sentence, not bracketed.

212 – ‘People in institutions’. These are not institutions. Write: ‘Clients of services catering to people with disabilities, drug addicts, the elderly, homeless, or women are 4.8 times more likely to experience mental health issues”.  

214 – Suggested re-write: “People experiencing alcohol, drug, or gambling addiction also show a very high OR ranked between 3 and 5.8”.

250 – Nothing is guaranteed. Rather; ‘Results show that a positive perceived quality of life is also a strong protector of mental health’

267 – bracketed text is unnecessary.

274 – It is not a debate. There are facts and consequences.

281 – Should be ‘Global Financial Crisis in 2008’

286 – Highlighted text should be a separate sentence.

450 – Incomplete reference

Author Response

To the International Journal of Environmental Research and Public Health editor,

Thank you for giving me the opportunity to submit a revised draft of my manuscript titled “Risk and protection factors of mental health conditions: Impact of employment, deprivation and social relationships” (retitled) to the International Journal of Environmental Research and Public Health.

We appreciate the time and effort that you and the reviewers have dedicated to providing your valuable feedback on our manuscript. We are grateful to the reviewers for their insightful comments on the paper.

Here is a point-by-point response to the reviewers’ comments and concerns. Also, we attached the manuscript with the subsequent corrections.

Thank you again for your support.

Reviewer 3

R3-1. I consider this manuscript acceptable for publication after some revisions. It needs professional editing to truncate excessive wording, which will illuminate the excellent content. Line 194 is an example,

“Results show that one of the sets that has the greatest impact on the presence of mental health conditions in households is the one linked to financial capacity”.

This can be truncated as:

“Results show that financial capacity is one of the greatest influences on mental health conditions in Spain”.

You may note my highlighted text where this has occurred elsewhere. There is also an excessive use of the phrase ‘in households’. Many of these can be simply deleted. See line 304 for example.

Author’s response 1: First of all, we would like to thank the reviewer for taking the time to review our manuscript, as all the comments and suggestions have improved the quality of the manuscript. We agree with the reviewer that further revision on English language and truncation of the phrases would be helpful. We tried to reformulate all the highlighted sentences on the text and tried to reduce the use of “in households”.

R3-2. Three other main changes are needed. First, the cited statistics on page 2 (lines 48 and 56) are outdated: the most recent Spanish statistics are important here. Second, I think Figure 1 can be removed as the statistics are unrelated to Spain.

Author’s response 2: We appreciate the reviewer insightful suggestion and agree that it would be useful to show representative data about the Spanish population for the work. However, for the initial framework, we think that it productive to start a minimal focus on the European context, for a later development of the Spanish data as in Table 1. Nevertheless, we agree that Figure 1 can be removed as its statistics has a very low relevance to the Spanish context.

Regarding the 2017 Health Survey, unfortunately they are the last published data by the Minister of Health. We are sorry that there are no more actualized data for the purpose and comparative of our research, we hope that a bigger focus on mental health research in Spanish research could emphasize the needs of more actualized the data by governmental agencies.

R3-3. Third, there needs to be a clear justification for using the Odds Ratio and Risk Ratio as well as a clear definition for each. Statistics in Table 2 need to be numerated as gross totals alongside the p and Odds Ratio columns. This will give the reader a clearer picture of the variables.

Author’s response 3: We apologize if our interpretation of OR was not included, as it clearly enhances the clarity of the methods. Also, we tried to avoid in the text the Risk Ratio terms, to refer only as the same statistical results and reduce variability of terms. The following clarification was added in Procedures and Analysis: “For clarification purposes, Odds ratios that are greater than 1 shall indicate that the event is more likely to occur as the predictor increases. Odds ratios that are less than 1 indicate that the event is less likely to occur as the predictor increases. This is translated interpretatively as protective factors on the protective model and risk factors on the risk model.”

Also, as Table 3 could also be confusing for the purpose of the analysis (as it could be interpreted as additional variables), we have erased it and added the following to Results: “Vital perception variables in Table 2 are related to the lack of life satisfaction and the perception of no improvement. In all cases, the OR is positive and greater than 1, reaching almost 2 in all the households in which the respondent did not feel satisfied with her/his life.”

R3-4. Some words are joined together because of a glitch (e.g., address the – line 37), which require separation. I’ve highlighted many in the text, but you’ll need to do a thorough survey of these.

Author’s response 4: We would like to thank the reviewer for pointing this out. We double checked the actual version of the manuscript with language experts to avoid these mistakes in its actual form.

R3-5. Do not use the terms ‘suffering’ or ‘problems’. This implies the person is a passive pawn to their irredeemable fate. Rather, use the words ‘manages’ and ‘issues’. For example, replace ‘individuals suffering from mental health problems’ with ‘people managing mental health issues’. Notice the difference? A person’s agency is emphasized.

Author’s response 5: Following the suggestion of the reviewer, both terms have been replaced by “mental ill-health” or “issues” instead of problems, and “manage” instead of “suffer” in the appropriate context on the manuscript, to emphasize the active aspect of the individuals in their own health.

R3-6.

Acronyms follow the name, which you’ve done correctly sometimes. You don’t need to use acronyms unless used frequently throughout the text (e.g., OR). Just state the name if only used once.   

36 - Delete paragraph. This is information incorrect and unsubstantiated.

123 – delete ‘n=’

134 – EINSFOESSA is not Survey on Social Needs and Integration. Cite the Spanish name if this is so.

146 – INE is not National Statistics Institute. Cite the Spanish name if this is so.

148 – delete (short questionnaire)

192 – The bracketed text should be a standalone sentence, not bracketed.

212 – ‘People in institutions’. These are not institutions. Write: ‘Clients of services catering to people with disabilities, drug addicts, the elderly, homeless, or women are 4.8 times more likely to experience mental health issues”.  

214 – Suggested re-write: “People experiencing alcohol, drug, or gambling addiction also show a very high OR ranked between 3 and 5.8”.

250 – Nothing is guaranteed. Rather; ‘Results show that a positive perceived quality of life is also a strong protector of mental health’

267 – bracketed text is unnecessary.

274 – It is not a debate. There are facts and consequences.

281 – Should be ‘Global Financial Crisis in 2008’

286 – Highlighted text should be a separate sentence.

450 – Incomplete reference

Author’s response 6: Following the suggestion of the reviewer, all these changes were made.

R3-7. 124 – Replace ‘%’ symbol with ‘percent’. The symbol is only used in tables or brackets.

Author’s response 7: As we received conflicting advice from a prior reviewer, we think it is better to maintain a minimum mathematical symbols among the text.

Reviewer 4 Report

General comments

As an exploratory study, I think this is a very interesting and necessary research to value the factors that can influence employment and mental health conditions. It focuses both on risk and protective factors, an approach that emphasize that both kinds of factors could be relevant to the integration in the labor market.

I believe this kind of research is important since it deeps on mental health and employment and emphasizes factors related to this topic.

It must be noted that in English the appropriate denomination for “protection” factors is PROTECTIVE factors.

Introduction

-The introduction is well constructed and coherent.

-The literature about the topic is justified and uses recent and related research’ findings.

-There is previous literature about the relations between social exclusion and poverty with mental health, but the approach taking into account risk and protective factors sounds to me interesting. Nevertheless, I would add some more literature about the different factors analyzed by the instrument.

-I don’t think the figure 1 is really necessary. Maybe with some synthetized data would be enough to portray the reality.

Methodology

Sample

-The study has a representative sample with 11655 households.

Instruments

The instrument is appropriate. It focuses specifically on the different areas of interest and deep on each factor.

Analyses

The analyses are appropriate. I was surprised about the possibility to use mental health as a dependent variable. Nevertheless, authors explain the relevance of this proposal analyzing, for example, how financial difficulties can limit the possibility to buy medicines or negative future outlooks can promote the onset of mental health.

References

References are mostly recent and well-presented.

Therefore, regarding the few possible changes needed in this research I would recommend its publication after minor revisions.

Author Response

To the International Journal of Environmental Research and Public Health editor,

Thank you for giving me the opportunity to submit a revised draft of my manuscript titled “Risk and protection factors of mental health conditions: Impact of employment, deprivation and social relationships” (retitled) to the International Journal of Environmental Research and Public Health.

We appreciate the time and effort that you and the reviewers have dedicated to providing your valuable feedback on our manuscript. We are grateful to the reviewers for their insightful comments on the paper.

Here is a point-by-point response to the reviewers’ comments and concerns. Also, we attached the manuscript with the subsequent corrections.

Thank you again for your support.

Reviewer 4

R4-1: As an exploratory study, I think this is a very interesting and necessary research to value the factors that can influence employment and mental health conditions. It focuses both on risk and protective factors, an approach that emphasize that both kinds of factors could be relevant to the integration in the labor market.

I believe this kind of research is important since it deeps on mental health and employment and emphasizes factors related to this topic.

Author’s response 1: We thank the reviewer for taking the time to assess our manuscript and the positive response to the theme of the manuscript as it inspires our research work.

R4-2: It must be noted that in English the appropriate denomination for “protection” factors is PROTECTIVE factors.

Author’s response 2: We thank the reviewer for pointing this out. “Protection factors” references were changed to “protective factors”.

R4-3: Introduction

-The introduction is well constructed and coherent.

-The literature about the topic is justified and uses recent and related research’ findings.

-There is previous literature about the relations between social exclusion and poverty with mental health, but the approach taking into account risk and protective factors sounds to me interesting. Nevertheless, I would add some more literature about the different factors analyzed by the instrument.

-I don’t think the figure 1 is really necessary. Maybe with some synthetized data would be enough to portray the reality.

Author’s response 3: We agree with the reviewer that Figure 1 did not match with the focus on the article, so we deleted it”.

R4-4: Methodology

-The study has a representative sample with 11655 households.

The instrument is appropriate. It focuses specifically on the different areas of interest and deep on each factor.

The analyses are appropriate. I was surprised about the possibility to use mental health as a dependent variable. Nevertheless, authors explain the relevance of this proposal analyzing, for example, how financial difficulties can limit the possibility to buy medicines or negative future outlooks can promote the onset of mental health.

References

References are mostly recent and well-presented.

Therefore, regarding the few possible changes needed in this research I would recommend its publication after minor revisions.

Author’s response 4: Thank you for your kind and encouraging words!